ET-1 receptor type B (ETBR) overexpression associated with ICAM-1 downregulation leads to inflammatory attenuation in experimental autoimmune myocarditis

Yang Peng 1
Li Fangfei 1
Tang Jiangfeng 1
Tian Qingshan 1
Zheng Zhenzhong 1 2 greateful@163.com
1 Cardiology, The First Affiliated Hospital of Nanchang University , Nanchang , China
2 Cardiology, Shenzhen Third People’s Hospital , Shenzhen, Guangdong , China
Uversky Vladimir
Electronic publication date: 2023 Oct 24
Publication date: 2023
Volume: 11
Electronic Location ID: e16320
Received 2023 May 23; Accepted 2023 Sep 29
Copyright: © 2023 Yang et al.
Copyright year: 2023
Copyright holder: Yang et al.
License: This is an open access article distributed under the terms of the Creative Commons Attribution License, which permits unrestricted use, distribution, reproduction and adaptation in any medium and for any purpose provided that it is properly attributed. For attribution, the original author(s), title, publication source (PeerJ) and either DOI or URL of the article must be cited.
License URL: https://creativecommons.org/licenses/by/4.0/

Keywords: ETBR, Autoimmune myocarditis, ICAM-1, Inflammation, Immune damage, Cardiac function, Overexpression, IFN-γ, IL-12, IL-17

Funding: 2020 Natural Science Foundation of Jiangxi Province in China 20202ABCL206002 This work was supported by grants from the 2020 Natural Science Foundation of Jiangxi Province in China (Project No.: 20202ABCL206002). The funders had no role in study design, data collection and analysis, decision to publish, or preparation of the manuscript.

==============================
Background

An experimental autoimmune myocarditis rat model was established by subcutaneous injection of porcine myocardial myosin (PCM). The effect of ET-1 receptor type B (ETBR) overexpression on autoimmune myocarditis was observed via tail vein injection of ETBR overexpression lentivirus in rats. We further investigated the mechanisms involved in the regulation of autoimmune myocarditis by ETBR overexpression.

Methods

Six rats were randomly selected from 24 male Lewis rats as the NC group, and the remaining 18 rats were injected with PCM on Day 0 and Day 7, to establish the experimental autoimmune myocarditis (EAM) rat model. The 18 rats initially immunized were randomly divided into three groups: the EAM group, ETBR-oe group, and GFP group. On Day 21 after the initial immunization of rats, cardiac echocardiography and serum brain natriuretic peptide (BNP) analysis were performed to evaluate cardiac function, myocardial tissue HE staining was performed to assess myocardial tissue inflammatory infiltration and the myocarditis score, and mRNA expression of IFN-γ, IL-12, and IL-17 was detected by qRT-PCR. Subsequently, immunohistochemical analysis was performed to detect the localization and expression of the ETBR and ICAM-1 proteins, and the expression of ETBR and ICAM-1 was verified by qRT-PCR and western blotting methods.

Results

On Day 21 after initial immunization, left ventricular end-diastolic diameter (LVEDd), left ventricular end-systolic diameter (LVEDs), and serum BNP concentrations increased in the hearts of rats in the EAM group compared with the NC group (P < 0.01), and ejection fraction (EF) and fractional shortening (FS) decreased compared with those of the normal control (NC) group (P < 0.01). LVEDd, LVEDs, and serum BNP concentrations decreased in the ETBR-oe group compared with the EAM group, while EF and FS increased significantly (P < 0.01). HE staining showed that a large number of inflammatory cell infiltrates, mainly lymphocytes, were observed in the EAM group, and the myocarditis score was significantly higher than that of the NC group (P < 0.01). Compared with that of the EAM group, myocardial tissue inflammatory cell infiltration was significantly reduced in the ETBR-oe group, and the myocarditis scores were significantly lower (P < 0.01). The mRNAs of the inflammatory factors IFN-γ, IL-12 and IL-17 in myocardial tissue of rats in the EAM group exhibited elevated levels compared with those of the NC group (P < 0.01) while the mRNAs of IFN-γ, IL-12 and IL-17 were significantly decreased in the ETBR-oe group compared with the EAM group (P < 0.01). Immunohistochemistry showed that the staining depth of ETBR protein in myocardial tissue was greater in the EAM group than in the NC group, and significantly greater in the ETBR-oe group than in the EAM group, while the staining depth of ICAM-1 was significantly greater in the EAM group than in the NC group, and significantly lower in the ETBR-oe group than in the EAM group. The ICAM-1 expression level was significantly higher in the EAM group than in the NC group (P < 0.01), and was significantly lower in the ETBR-oe groupthan in the EAM group (P < 0.01).

Introduction

Myocarditis is an inflammatory disease of the heart muscle with various etiologies, with viral infections being the most common (Sagar, Liu & Cooper, 2012). The pathogenesis of viral myocarditis includes direct damage to the myocardium by viral infection and indirect damage caused by the host immune response (Zhao & Fu, 2018). There is increasing evidence that overactive inflammatory and autoimmune responses triggered by viral infections, rather than direct viral infections, are the main cause of the development of myocarditis (Pan et al., 2019; Reddy et al., 2013; Yajima & Knowlton, 2009). Approximately 40–66% of patients with myocarditis were reported to recover completely on their own within the first 4–12 weeks, but approximately half of patients with acute myocarditis with autoimmune myocardial injury exhibit major signs of heart failure, arrhythmias and sudden cardiac death (Błyszczuk, 2019). Unfortunately, the mechanisms of myocarditis have not been fully elucidated and there are no effective treatment strategies for myocarditis, which makes myocarditis a serious cardiovascular health problem.

Endothelin-1 (ET-1) is a 21-amino acid active peptide, that is the main vasoconstrictor secreted by endothelial cells and has pro-oxidant and inflammatory effects (Iglarz & Clozel, 2007). ET-1 mediates its action through two types of receptors, ET-1 receptor type A (ETAR) and ET-1 receptor type B (ETBR). Under physiological conditions, ETAR is expressed in smooth muscle cells and can mediate vasoconstriction, whereas ETBR is mainly found in endothelial cells and mediates vasodilation (Iglarz et al., 2015). Current studies have shown that the endothelin system is widely involved in cardiovascular diseases and that ETBR is closely associated with pulmonary hypertension, heart failure, artery hypertension, atherosclerosis, and chronic kidney disease (Halcox et al., 2007; Honoré et al., 2005; Seccia & Calò, 2017; Tabeling et al., 2022; Valero-Munoz et al., 2016). However, there are fewer studies exploring the correlation between ETBR and inflammation because most attention has been focused on the vasodilatory function of ETBR, and ETBR has not been reported in myocarditis.

Intercellular adhesion molecule-1 (ICAM-1) is a cell surface glycoprotein and adhesion receptor that regulates the recruitment of leukocytes from the circulation to inflammatory sites (Bui, Wiesolek & Sumagin, 2020). ICAM-1 is mainly expressed in immune cells, endothelial cells and epithelial cells, and a variety of inflammatory factors highly induce ICAM-1 expression, suggesting that ICAM-1 plays an important role in mediating immune and inflammatory responses (Yusuf-Makagiansar et al., 2002). Liu, Li & Zhao (2022) found that curcumin attenuated LPS-induced myocarditis in vitro by inhibiting ICAM-1/CD40/NF-κB. Previous studies have found that ICAM-1 expression is also regulated by some microRNAs. MicroRNA-141 was found to downregulate ICAM-1 in endothelial cells, thereby reducing leukocyte adhesion and attenuating myocardial ischemia-reperfusion injury (Liu et al., 2015). Furthermore, microRNA-27 can target ICAM-1 and protect against LPS-mediated inflammatory injury in H9c2 cells by inhibiting ICAM-1 expression (Anzai et al., 2019). Therefore, inhibition of ICAM-1 production and release may be a potential therapeutic strategy for myocarditis.

The correlation between ETBR and ICAM-1 has already been reported. Buckanovich et al. (2008) showed that ETBR blockade increases T-cell homing to tumors, and increases ICAM-1 mRNA and protein expression via NO suppression. Moreover, Granström et al. (2004) showed that ETBR expression was increased in bronchial smooth muscle cells in an experimental airway inflammation model, suggesting that ETBR may be involved in the inflammatory process. Hence, we hypothesized that ETBR overexpression may lead to inflammatory attenuation in experimental autoimmune myocarditis in rats, which is associated with the protective effect of ICAM-1 downregulation. Based on the above, we first observed the effects of ETBR on cardiac function and inflammation in rats with myocarditis using an experimental autoimmune myocarditis (EAM) rat model. The expression level of ICAM-1 in the ETBR-overexpressing EAM model was further examined to verify the regulation of ICAM-1 expression by ETBR.

Materials and Methods

Animals

Twenty-four male Lewis rats (age, 6–7 weeks), weighing 180–200 g, were purchased from Vital River Laboratory Animal Technology Co. (Beijing, China). Animals were kept under standard conditions with a mean temperature of 21 ± 2 °C, a mean relative humidity of 50% ±20% and a defined day-and-night-cycle of 12 h light and 12 h dark. No control diet was required in this experiment, and all diets were standardized. All animals were euthanized before the end of the experimental program. The details of the method are described below. Rats were anesthetized with an intravenous overdose of pentobarbital (100–150 mg/kg) and killed by cervical dislocation under deep anesthesia. At the scene of execution, other animals should not be present, and the carcass should be properly disposed only after the death of the animal is confirmed. The experiments were conducted in accordance with the “Guide for the Care and Use of Laboratory Animals” published by the United States National Institute of Health (Publication No. 85-23, revised in 1996), and all the performed experiments were approved by the Ethics Committee of the Animal Care and Use Committee of The Affiliated Hospital of Nanchang University, Nanchang, China (IACUC Issue No: 202205QR004).

Construction of the recombinant lentiviral vector

The gene sequence of rat ETBR (NM_017333) was first obtained by searching the NCBI website, and the target gene sequence (Forward: 5′- GGAATTGGGGTTCCAAAATG-3′; Reverse: 5′- CCTTATAGTCCTTATCATCGTC -3′) was synthesized by chemical synthesis. The recombinant positive clones were selected for PCR and sequencing, and the lentiviral vector containing ETBR-shRNA was obtained by ligation with the GV492 vector. The instrumental vector plasmid carrying the ETBR gene sequence and the viral packaging helper plasmid were cotransfected into 293T cells, and the virus was concentrated and purified at 35,000 rpm for 100 min using ultracentrifugation at 48–72 h after the completion of transfection to finally obtain the complete ETBR gene overexpression lentiviral vector. The ETBR protein was detected by Western blots to confirm whether the ETBR gene overexpression lentiviral vector was successfully packaged. The negative control virus was a green fluorescent protein (GFP) null-loaded lentivirus. The viral titer was determined based on the expression of GFP, and the titer of the viral strain was routinely 1 × 108 TU/ml.

Establishment of EAM rat models

The EAM rat model was constructed as described in our previous study (Zheng et al., 2018). Briefly, purified porcine myocardial myosin (PCM; Sigma Aldrich, St. Louis, MI, USA) was dissolved in 0.15 mol/l phosphate buffered saline (PBS) and the final concentration was adjusted to 2 g/l. Then the myosin was emulsified with Complete Freud’s Adjuvant (CFA) (Sigma Aldrich, St. Louis, MI, USA) in a 1:1 ratio. Each rat in the experimental group was injected subcutaneously with 200 μl of PCM-CFA emulsion in the inguinal region, foot pad region, and axilla on Day 1 and Day 7. The rats in the control group were injected subcutaneously with 100 μl of CFA on Day 0 and Day 7.

Experimental study design and grouping

Six normal rats were randomly selected as the normal control group (NC group), and the remaining 18 rats were injected with PCM on Days 0 and 7 to establish the EAM rat model. The 18 rats were randomly divided into three groups: EAM group, ETBR-oe group and GFP group. The ETBR gene overexpression lentivirus was removed from the −80 °C freezer and the lentiviral reagent was diluted with saline. Every rat in the ETBR-oe group was injected with a 200 μl dilution of ETBR gene overexpression lentivirus (containing 1 × 108 TU lentivirus) into the tail vein on the day of the initial immunization, while the rats in the GFP group were injected with the same dose of empty lentivirus (GFP-LV) and the rats in the NC and EAM groups were injected with the same dose of saline.

Assessment of cardiac function by echocardiography

On Day 21 after the initial immunization, cardiac echocardiography was performed in each group of rats to evaluate cardiac structure and cardiac function. The anesthesia machine and ultrasound system were connected, and the concentration of anesthetic gas isoflurane was adjusted to a suitable value to confirm that the rats were well anesthetized. Parasternal left ventricular long-axis views were taken, M-section images were obtained at the level of the mitral tendon cords, and the left ventricular end-diastolic diameter (LVEDd), left ventricular end-systolic diameter (LVEDs), left ventricular fractional shortening (FS), left ventricular ejection fraction (EF), left ventricular end-diastolic posterior wall thickness (LVPWd), and left ventricular end-systolic posterior wall thickness (LVPWs) were measured. A minimum of five cardiac cycles were measured in each rat.

Histopathological analysis

Myocardial tissue was fixed in 10% formaldehyde solution and embedded in solid paraffin at 65–70 °C. The thickness of the sections was fixed at 2–3 μm and stained with hematoxylin and eosin (HE). The pathological changes in myocardial tissue such as cardiomyocytes, myocardial interstitium, and inflammatory cells were observed under a light microscope. The percentage of the area of inflammatory cell infiltration and necrotic area of myocardial tissue in each field of view to the whole field of view was calculated, and its average value was used for myocarditis scores. The myocarditis scores were exactly as follows: 0, no inflammation; 1, <25% of the involved heart portion; 2, 25–50%; 3, 50–75%; and 4, >75% (Tajiri et al., 2012). Data were analyzed by an observer who was blinded to the treatment of the rats.

Immunohistochemical analysis

To detect the expression and distribution of ETBR and ICAM-1 in myocardial tissue, we performed immunohistochemical analysis on prepared paraffin sections of myocardial tissue. Paraffin sections were baked at 65 °C for 2 h. After dewaxing and rehydration, antigen repair was performed on tissue sections in citrate buffer at pH 6.0. PBS was used to wash the sections, and then endogenous peroxidase was removed with 3% H2O2 for 30 min. A total of 50 μl of normal goat serum was added to each section for closure and incubated for 10 min at room temperature. The protein primary antibody was incubated in a wet box overnight at 4 °C in the refrigerator, followed by washing the sections and incubating the secondary antibody for 50 min at 37 °C. The signal was amplified with an acid-biotin-horseradish peroxidase procedure, using diaminobenzidine as a color developer. The semiquantitative levels and localization of ETBR and ICAM-1 proteins were observed microscopically. Moreover, in order to detect the expression of ETBR in myocardial tissue, we co-stained ETBR with HE staining performed as described previously by Grosset et al. (2019).

Enzyme‑linked immunosorbent assay (ELISA) for serum brain natriuretic peptide (BNP)

After echocardiography, blood was collected from the inferior vena cava of the rats. Venous blood was centrifuged for 10 min and rat serum was obtained. The serum BNP concentration was measured by a BNP assay kit (Cloud-Clone, cat. No. CEA541Ra) according to the ELISA kit specification.

Quantitative real-time reverse transcription polymerase chain reaction (qRT-PCR)

RNA was extracted and isolated from rat myocardial tissue by homogenization, centrifugation and dissolution. A PCR tube was taken, a solution containing more than 100 ng of RNA was added as a template for reverse transcription, and 1 µl of reverse transcription primer was added for reverse transcription. This process was performed according to the instructions of RevertAid First Strand cDNA Synthesis Kit from TransGen Biotech Co. (Beijing, China). qRT-PCR analysis was then performed using the ABI Prism 7000 system (Abcam, Fremont, CA, USA) to detect mRNA expression. The relative mRNA expression levels of each molecule were normalized by subtracting the corresponding GAPDH threshold cycle (CT), which was performed using the ΔΔCT comparison method. A list of real-time PCR primer sequences is presented in Table 1.

Table 1 Primers used for QRT-PCR.

Primer name	Primer sequences (5′-3′)	
ETBR-F	CCTTTTGTCCGAGCCAGAGC	
ETBR-R	GGATTGGAAGCACCAGGAGAA	
ICAM-1-F	CTGTCGGTGCTCAGGTATCC	
ICAM-1-R	TGTCTTCCCCAATGTCGCTC	
GAPDH-F	CAAGTTCAACGGCACAGTCAAG	
GAPDH-R	ACATACTCAGCACCAGCATCAC	
IFNγ-F	TCCTCTTTGACCAATCATTCTTTCT	
IFNγ-R	ATTCCTCTGGTCAGCAGCAC	
IL-12-F	TGACATGTGGACGAGCATCT	
IL-12-R	CAGTTCAATGGGCAGGGTCT	
IL-17-F	AAACGCCGAGGCCAATAACT	
IL-17-R	GGTTGAGGTAGTCTGAGGGC	

Western blot analysis

Total proteins from myocardial tissue were extracted by radioimmunoprecipitation assay (RIPA) lysis buffer (cat. No. R0020; Solarbio, Beijing, China). Protein was quantified using the BCA protein assay kit (cat. No. P0012; Beyotime, Beijing, China). Proteins (40 ug) were separated by 10% sodium dodecyl sulfate‒polyacrylamide gel electrophoresis (SDS-PAGE) (Bio-Rad, Hercules, CA, USA) and transferred to nitrocellulose filter membranes (Millipore, Billerca, MA, USA) for 100 min at 300 mA. Then, the membranes were blocked for 1.5 h at room temperature with 5% skimmed milk or 3% bovine serum albumin (BSA) in Tris-buffered saline containing 0.05% Tween (TBS-T). The membranes were incubated with primary antibodies against ETBR (cat. No. ab262700, 1:1,000; Abcam, Fremont, CA, USA), ICAM-1 (cat. No. ab222736, 1:1,000; Abcam, Fremont, CA, USA), and GAPDH (cat. No. ab37168, 1:1,000; Abcam, Fremont, CA, USA) at 4 °C overnight and with a secondary antibody (cat. No. SA00001-2, 1:5,000; Proteintech, Rosemont, IL, USA) at room temperature for 1 h. The proteins on the membranes were visualized using an enhanced chemiluminescence (ECL) kit (cat. No. PK10003; Proteintech, Rosemont, IL, USA). The exposed protein bands were converted to data using Image software (Bio-Rad, Hercules, CA, USA) to obtain grayscale values for result analysis.

Statistical analysis

SPSS version 20.0 software (IBM Corporation, Armonk, NY, USA) was used to analyze the statistical data obtained from the experiments, and the data are presented as the mean ± SEM. GraphPad Prism version 8.0.1 software (GraphPad Software, Inc, San Diego, CA, USA) was used to produce statistical graphs. Ordinal logit regression was used for ordinal and noncontinuous data with more than two groups while one-way ANOVA was used for continuous data with more than two groups. A P value < 0. 05 was considered statistically significant (*P < 0.05, **P < 0.01).

Results

Measurement of body weight

Before the EAM model induction by PCM, the body weight of all rats was 191.5 ± 4.2 g, and there was no significant difference in the body weight of rats in each group before immunization. Compared with the rats in the NC group, the rats in the EAM group and the GFP group showed a dramatic decrease in body weight. However, the rats in the ETBR-oe group had significantly higher body weights than the rats in the EAM group, suggesting that ETBR overexpression reversed this effect (Fig. 1).

Figure 1 Measurement of body weight.

Data were analyzed by analysis of variance (*P < 0.05 vs NC group; #P < 0.05 vs EAM group and GFP group).

Measurement of ETBR expression in myocardial tissues by qRT-PCR and western blot analysis

To explore the effect of ETBR overexpression on autoimmune myocarditis, we first infected rats with EAM with lentivirus carrying the ETBR gene. qRT-PCR detected the relative mRNA level of ETBR, and the results showed that the ETBR mRNA expression level in the ETBR-oe group was significantly higher than that in the EAM and NC groups (Fig. 2A). Moreover, western blotting results showed that the protein expression level of ETBR was significantly higher in the ETBR-oe group than in the EAM and GFP groups (Fig. 2B), indicating that lentivirus infection was successful.

Figure 2 Measurement of ETBR expression in myocardial tissues after lentiviral transfection.

(A) The relative mRNA level of ETBR detected by QRT-PCR method. (B) The protein expression level of ETBR detected by western blotting assay. Data were analyzed by analysis of variance (*P < 0.05, **P < 0.01).

Effects of ETBR overexpression on cardiac function in rats with EAM

Echocardiography was performed on Day 21 after PCM immunization in rats to assess the effect of ETBR overexpression on cardiac function. Compared with those of the NC group, LVEDd and LVEDs were significantly increased (Figs. 3A and 3B), while EF and FS were significantly decreased in the EAM group rats (Figs. 3C and 3D). In addition, compared with those of the EAM group, the LVEDd and LVEDs of the rats in the ETBR-oe group were significantly decreased while the EF and FS were significantly increased, suggesting that ETBR gene overexpression could alleviate ventricular dilation and improve cardiac function in rats with EAM. To explore the effect of ETBR overexpression on the afterload of the left ventricle, we further examined the posterior wall thickness of the left ventricle. It was found that LVPWd and LVPWs were elevated in the rats in the EAM group compared with the control group. However, LVPWd and LVPWs were significantly lower in the rats in the ETBR overexpression group than in the rats in the EAM group (Figs. 3E and 3F).

Figure 3 Effects of ETBR overexpression on cardiac function in EAM rats.

(A) LVEDd of rats in each group was measured by echocardiography; (B) LVEDs of rats in each group was measured by echocardiography; (C) EF of rats in each group was measured by echocardiography; (D) FS of rats in each group was measured by echocardiography; (E) LVPWd of rats in each group was measured by echocardiography; (F) LVPWs of rats in each group was measured by echocardiography. Data were analyzed by analysis of variance (*P < 0.05, **P < 0.01).

Effects of ETBR overexpression on inflammatory cell infiltration in rats with EAM

On the Day 21 after the initial immunization, HE staining showed that the cardiomyocytes in the NC group were neatly arranged and morphologically normal, and no inflammatory cell infiltration was observed in the interstitium (Fig. 4A, left upper panel). In contrast, in the EAM group, myocardial cells were disorganized, some of them were lysed, inflammatory cells were evidently found in the interstitial part (Fig. 4A, right upper panel), and the myocarditis score was significantly higher than that of the NC group (P < 0.01) (Fig. 4B). In the GFP group, cardiomyocytes were disorganized with some degree of inflammatory cell infiltration (Fig. 4A, right lower panel), and the difference in the myocarditis score was not statistically significant compared with that in the EAM group (P > 0.05). In the ETBR-oe group, only a small number of cardiomyocytes were disorganized and swollen, inflammatory cell infiltration was significantly reduced compared with that in the EAM group (Fig. 4A, left lower panel), and the myocarditis score was significantly lower than that in the EAM group (P < 0.01). Therefore, ETBR overexpression alleviates inflammatory cell infiltration in rats with EAM.

Figure 4 Effects of ETBR overexpression on inflammatory cell infiltration in EAM rats.

(A) HE staining for detection of inflammatory cell infiltration in myocardial tissue. (B) Myocarditis score for detecting inflammatory levels in myocardial tissue. Data were analyzed by analysis of ordinal logit regression (**P < 0.01).

Effects of ETBR overexpression on serum BNP concentration

The relative serum BNP concentration was significantly higher in the EAM group than in the NC group on Day 21 after the initial immunization (P < 0.05). The relative serum BNP concentration in the GFP group was not significantly different from that in the EAM group (P > 0.05). However, the relative serum BNP concentration was significantly lower in the ETBR-oe group than in the EAM group (P < 0.05) (Fig. 5).

Figure 5 Effects of ETBR overexpression on serum BNP concentration.

The relative serum BNP concentration was detected by ELISA analysis. Data were analyzed by analysis of variance (**P < 0.01).

Effects of ETBR overexpression on the expression of the inflammatory cytokines IFN-γ, IL-12 , and IL-17

The relative mRNA levels of the inflammatory factors IFN-γ, IL-12 and IL-17 in the myocardial tissue of the rats in the EAM group were elevated compared with those in the NC group, and the differences were significant (P < 0.01). The relative mRNA levels of IFN-γ, IL-12 and IL-17 were significantly decreased in the ETBR-oe group compared with the EAM group (P < 0.05) (Fig. 6).

Figure 6 Effects of ETBR overexpression on expression of inflammatory cytokines IFN-γ, IL-12 and IL-17.

QRT-PCR detected the relative mRNA level of IFN-γ, IL-12 and IL-17. Data were analyzed by analysis of variance (*P < 0.05, **P < 0.01).

Distribution and expression of the ETBR and ICAM-1 proteins in myocardial tissue by immunohistochemical assays

To investigate the relationship between ETBR and ICAM-1, we first reviewed and collected paraffin sections of myocardial tissues from each group of rats, and then used immunohistochemistry to detect the distribution of ETBR and ICAM-1 in the myocardial tissues of rats with EAM. Immunohistochemical results showed that ETBR protein was mainly localized in the myocardial cell membrane and cell plasma. Moreover, ETBR protein staining was significantly more intense in myocardial tissues in the ETBR-oe group than in the EAM group, whereas ICAM-1 staining was significantly less intense in myocardial tissues in the ETBR-oe group. These data suggest that both ETBR and ICAM-1 were expressed in myocardial tissues and that ETBR overexpression may inhibit the expression of ICAM-1 in the EAM model, which needs to be further verified using western blotting (Fig. 7).

Figure 7 Distribution of ETBR and ICAM-1 proteins in myocardial tissue by immunohistochemical assay.

The brownish-yellow particles are ETBR and ICAM-1 proteins, and the blue color is the nucleus. The more specific proteins are expressed, the darker the brownish-yellow particles are.

Effects of ETBR overexpression on ICAM-1 expression in rats with EAM

To confirm the effect of ETBR overexpression on ICAM-1 expression in EAM rat models, we detected the relative expression levels of ICAM-1 mRNA in the myocardial tissue of each group by qRT-PCR and western blotting methods. The results revealed that the mRNA level of ICAM-1 in myocardial tissues of the EAM group was significantly elevated, while the ICAM-1 mRNA level was significantly reduced after ETBR overexpression (Fig. 8A). In addition, the ICAM-1 protein expression level was significantly elevated in the myocardial tissues of the rats in the EAM group compared with those in the NC group, and ETBR overexpression reversed this effect (Fig. 8B).

Figure 8 Effects of ETBR overexpression on ICAM-1 expression in EAM rats.

(A) The relative mRNA level of ICAM-1 detected by QRT-PCR method. (B and C) The protein expression level of ICAM-1 detected by western blotting assay. Data were analyzed by analysis of variance (*P < 0.05, **P < 0.01).

Discussion

Myocarditis is an inflammatory disease of the heart muscle caused by various etiologies, mainly viral infection and postinfection autoimmunity, often developing into dilated cardiomyopathy and heart failure (Basso, 2022). However, the mechanism by which immune damage leads to cardiac dysfunction and heart failure remains unclear. EAM is an animal model that mimics myocarditis and has become an important tool for understanding the mechanisms of immune damage in myocarditis. In this experiment, the EAM model was induced by subcutaneous injection of PCM, and the mRNA and protein expression levels of ETBR were found to be significantly increased in the myocardial tissue of rats in the EAM model group. Therefore, ETBR may play a key role in the immune damage of autoimmune myocarditis.

To clarify the effect of ETBR overexpression on EAM, we first prepared an ETBR overexpression lentivirus and then intervened in autoimmune myocarditis by injecting an ETBR overexpression lentivirus. We found that LVEDd and LVEDs decreased, while EF and FS increased significantly in the ETBR-oe group compared with the EAM group, indicating an improvement in cardiac function. Serum BNP concentration is a key biomarker for the diagnosis of heart failure with good sensitivity (Santaguida et al., 2014). We detected the BNP concentration in serum and found that the results of serological examination and echocardiography were consistent. Serum BNP was also significantly lower in the rats in the ETBR-oe group than in the EAM group. Moreover, this experiment further explored the effect of ETBR overexpression on inflammation levels in rats with EAM. We found that after ETBR overexpression intervention in the EAM model, myocardial tissue showed only a small amount of myocardial cell arrangement disorder and cell swelling, inflammatory cell infiltration was significantly reduced, and the myocarditis score was significantly decreased, suggesting that ETBR overexpression can reduce the inflammatory pathological damage of autoimmune myocarditis.

Autoimmune myocarditis is considered immunologically to be a CD4+ T lymphocyte-mediated immune damaging disease (Anzai et al., 2019). CD4+ T cells can be classified into different effector subpopulations based on their biological functions and specific cytokine production, mainly Th1, Th2, and Th17 cells (Ruterbusch et al., 2020). Th1 cells, which secrete mainly IL-2, IFN-γ and IL-12, and Th17 cells, which secrete mainly IL-17, exhibit proinflammatory properties in myocarditis (Chang et al., 2019; McCarthy et al., 2015). The cooperation of Th1 and Th17 cells determines the transition from autoimmune myocarditis to dilated cardiomyopathy (Nindl et al., 2012). Zhang et al. (2016) found that apigenin attenuated experimental autoimmune myocarditis in mice by reducing Th1-related inflammatory cytokines (IFN-γ and IL-2). Moreover, Su et al. (2011) found that inhibition of IL-17 secreted by Th17 cells reduced the severity of myocarditis and improved the cardiac pathological changes of myocarditis. We examined IFN-γ, IL-12, and IL-17 cytokines and found that the mRNA levels of the inflammatory factors IFN-γ, IL-12 and IL-17 in the myocardial tissue of rats in the EAM group were elevated compared with those in the NC group. The results suggested that Th1 and Thl7 cells were active in rats with EAM. We subsequently intervened rats with EAM by injecting an ETBR overexpression lentivirus, and the results suggested that ETBR overexpression decreased the mRNA expression levels of Th1- and Th17-related cytokines.

Previous studies have shown that ICAM-1 plays an important role in mediating immune and inflammatory responses. In the study by Luo et al. (2022), monocytes enhanced the inflammatory response to promote calcific aortic valve disease progression through a β2 integrin/ICAM-1 mediated signaling pathway. Moreover, Ho et al. (2023) found that miR-146a reduces vascular inflammatory responses through inhibition of ICAM-1 expression. However, the role of ICAM-1 in myocarditis has not been clarified. We found that both ETBR and ICAM-1 were elevated in the myocardial tissue of rats with EAM, and ICAM-1 expression levels were significantly decreased after ETBR overexpression. Therefore, ETBR overexpression to alleviate autoimmune myocarditis may be associated with ICAM-1 downregulation.

In summary, our study first demonstrated that ETBR overexpression reduced inflammation and improve cardiac function in rats with EAM, which may be associated with the downregulation of ICAM-1. However, our study has several limitations. We did not validate the interaction between ETBR and ICAM-1 by immunoprecipitation, and did not detected the expression of ETBR and ICAM-1 in patients with myocarditis. Previous animal studies have shown that EAM is an autoimmune disease mainly mediated by CD4+ T lymphocytes but not CD8+ T lymphocytes (Vdovenko & Eriksson, 2018). CD4+ T lymphocyte homing is considered the initiating event in autoimmune myocarditis, which is highly dependent on ICAM-1 (Deane et al., 2012; Tanaka et al., 2011). Further studies are warranted to investigate the role of ICAM-1 in cardiac homing of CD4+ T lymphocytes during the pathological process of autoimmune myocarditis.

Conclusion

ETBR overexpression significantly reduced inflammatory levels and pathological changes, and significantly increased cardiac function in rats with EAM, probably due to the downregulation of ICAM-1. Thus, ETBR may be a promising novel therapeutic target for myocarditis.

Supplemental Information

Supplemental Information 1 Author checklist.

Click here for additional data file.

Supplemental Information 2 Raw data exported from body weight of rats in each group, validation of successful ETBR lentiviral overexpression, and effects of ETBR overexpression on cardiac function for Figures 1–3.

Click here for additional data file.

Supplemental Information 3 Raw data exported from effects of ETBR overexpression on inflammatory cell infiltration detected by histopathological and immunohistochemical analysis for Figure 4.

Click here for additional data file.

Supplemental Information 4 Raw data exported from expression and distribution of ETBR and ICAM-1 proteins in myocardial tissue by immunohistochemical assay for Figure 7.

Click here for additional data file.

Supplemental Information 5 Raw data exported from effects of ETBR overexpression on serum BNP concentration, ETBR overexpression on expression of inflammatory cytokines IFN-γ, IL-12 and IL-17, and ETBR overexpression on ICAM-1 expression for Figures 5, 6, and 8.

Click here for additional data file.

Abbreviation

ET-1 Endothelin-1

ETBR Endothelin-1 receptor type B

ICAM-1 Intercellular adhesion molecule-1

EAM Experimental autoimmune myocarditis

PCM Porcine myocardial myosin

qRT-PCR Quantitative real-time reverse transcription polymerase chain reaction

LVEDd Left ventricular end-diastolic diameter

LVEDs Left ventricular end-systolic diameter

EF Left ventricular ejection fraction

FS Left ventricular fractional shortening

LVPWd Left ventricular end-diastolic posterior wall thickness

LVPWs Left ventricular end-systolic posterior wall thickness

BNP Brain natriuretic peptide

Additional Information and Declarations

Competing Interests

Author Contributions

Animal Ethics

Data Availability

The authors declare that they have no competing interests.

Peng Yang conceived and designed the experiments, performed the experiments, analyzed the data, prepared figures and/or tables, authored or reviewed drafts of the article, and approved the final draft.

Fangfei Li analyzed the data, prepared figures and/or tables, and approved the final draft.

Jiangfeng Tang analyzed the data, prepared figures and/or tables, and approved the final draft.

Qingshan Tian analyzed the data, prepared figures and/or tables, and approved the final draft.

Zhenzhong Zheng conceived and designed the experiments, performed the experiments, authored or reviewed drafts of the article, and approved the final draft.

The following information was supplied relating to ethical approvals (i.e., approving body and any reference numbers):

The Ethics Committee of the Animal Care and Use Committee of The Affiliated Hospital of Nanchang University, Nanchang, China (202205QR004).

The following information was supplied regarding data availability:

The raw measurements are available in the Supplemental Files.

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
