# Peer review of "ET-1 receptor type B (ETBR) overexpression associated with ICAM-1 downregulation leads to inflammatory attenuation in experimental autoimmune myocarditis"

_PeerJ, doi:10.7717/peerj.16320_

## Round 0.1 · original submission · Major Revisions

Please address the issues pointed out by the reviewers and amend the manuscript accordingly.

·

Basic reporting

Summary
In this manuscript, the author investigates the molecular mechanism of how Endothelin B Receptor (ETbR) reduces myocarditis severity in Porcine myocardial myosin (PCM)-induced autoimmune endocarditis rats model.
The author successfully set up a rat model that overexpresses ETbR and then induces myocarditis by injecting these rats with PCM. First, the author showed that rats with ETbR overexpression preserve cardiac function as measured by echocardiogram. At the histology level, ETbR overexpression leads to significantly less lymphocyte infiltration in cardiomyocytes than in wild-type PCM-induced rats. The author then measured the ICAM-1 expression level in cardiomyocytes and showed that rats with ETbR overexpression had reduced ICAM-1 expression compared to wild-type rats with PCM induction. The author concluded that ETbR overexpression is associated with ICAM-1 downregulation, reducing lymphocyte infiltration and inflammation in cardiomyocytes.

Basic Reporting
Overall the manuscript is easy to read and easy to understand. The English language is relatively good, with minimal need for editorial edits. The manuscript is self-contained, and the story is complete in itself.
The literature background is relatively good except in the introduction section. I wonder why the author chose to study ETbR as a modulator of inflammation in myocarditis. I can see why ETbR upregulation might be related to a better outcome of myocarditis since it could potentially reduce cardiac afterload. However, I find it challenging to relate ETbR expression with the degree of inflammation. Is there some transcriptome/proteome study that shows ETbR is associated with better clinical outcomes of myocarditis? I also find connecting ETbR and ICAM-1 challenging since ETbR is a receptor, while ICAM-1 is a surface marker. What kind of evidence in the literature makes the author think these two pathways are related? Do they share the same signaling molecules or pathways?
The figures still need some edits.
My primary concerns for this section are:
1. Page 9, Lines 110 – 112, the author mentions the siRNA experiment, but no knockdown investigation exists in this manuscript. Is this intentional?

Experimental design

Experimental Design
Overall, the experimental design is logical and easy to understand. However, I find that there are some details of the methodology that is still missing. I have listed these comments below.
1. Could the author add the source and catalog number of antibodies used in this study, e.g., anti-ICAM1, anti-ETbR, anti-GFP, and anti-GAPDH
2. Page 12, Lines 217 – 219: What are EABR I and EABR II antibodies? Does the author mean ETBR primary and secondary antibodies?
3. Line 136 – 138: How was the virus concentrated and purified?
4. Line 151 – 157: was the virus suspended in saline, not HEK293 media?

Validity of the findings

Validity of the findings
The finding of this manuscript is interesting. However, the quality of the figure and presentation could be improved. Also, some data are missing to make a convincing case that ETbR overexpression is associated with inflammation reduction of myocarditis.
The discussion part is well-written and well-explained. The author correctly states that the direct relationship between ETbR and ICAM-1 in this manuscript is not fully validated. However, the author did not claim any direct interaction, to begin with. So this part of the claim is still valid.
Major Concerns
1. I would not use the word 'inhibiting' in the title since it implies direct interaction. I would soften the phrase to 'ETBR overexpression associates with ICAM-1 downregulation leads to inflammation attenuation of experimental autoimmune myocarditis in rats'
2. Figure 1B: Could the author show the Western blot (WB) with ETbR, GFP, and GAPDH bands on the same image with the ladder?
Also, I checked supplementary file 2, which contains WB images. I find 3 files of ETbR blot and 2 files of GAPDH blot. However, the protein lanes in those images don't match. Specifically, in ETbR blots, the image has 4 bands, then a ladder, then another 4 bands. In contrast, GAPDH WB had 8 bands connected. Were these blots run in the same gel or separately? Could the author label what those bands of WB are in the supplementary file 2.
3. In the cardiac function study, could the author provide videos of the echocardiogram performed in each experimental group? I would like to see the cardiac movement myself.
4. Since ETbR is known to mediate vasodilation, is ETbR overexpression possibly leading to better cardiac function in PCM treated group because of reduced cardiac afterload? One way that the author could check this effect is to measure tail-cuff pressure by comparing between ETbR-overexpressed group to the negative control group. Does the author have this data to share with us?
5. Can the author include general health data of the rats, such as body weight, activity, and amount of feed that the rats take daily? I wonder if overexpression of ETbR has any side effect on overall rats' health.
6. In Figure 2A: Can the author co-stain ETbR with H&E in these tissue as shown in this method paper (Grosset et al., 2019)?
7. The author should mention in a few sentences why BNP measurement is necessary. I know it is a marker of heart failure, but some audiences might not know its significance.
8. Similar to point 6, the author should mention why IL17, IFN-g, and IL12 measurement is necessary in a few sentences.
9. Figure 5: the author needs a more quantitative measurement to show the anti-correlation between ETbR and ICAM-1 expression. Western blot is a good start. Another experiment the author can do is immunofluorescence co-staining of ETbR and ICAM-1 in all experiment groups.
10. Figure 5: in IHC for ETbR in the ETbR-Overexpression group, why are there so few muscle fibers in that image compared to that of the image below it or Figure 2A?
11. In the discussion section, the author should discuss the possible pathway that could link ETBR and ICAM-1 together so that further study can follow up the molecular mechanism of this finding.
Minor concerns
1. Figure 1B: the fonts of the bar chart label are distorted. Can the author fix the aspect ratio?
2. Figure 1B: Can I see the GFP band in this Western blot as well
3. Table 2: The cardiac function should be presented in bar chart format.
4. The figure reference in the text, lines 250 – 261, should be referenced at the end of this sentence to help the audience follow the story. For example :
On day 21 after the initial immunization, H&E staining showed that the cardiomyocytes in the NC group were neatly arranged and morphologically normal, and no inflammatory cell infiltration was seen in the interstitium (Figure 2A, left upper panel).
5. Figure 2A: the fonts on the figure are distorted. Can the author fix the aspect ratio?
6. Figure 2B: The statistics for this data should be ordinal logit regression, not ANOVA since the score is ordinal and non-continuous by nature. But it won't change the conclusion of this graph. So it is a minor comment.
7. Line 341-342: I would avoid using the word inhibit. I would state that ETbR overexpression is associated with the downregulation of ICAM-1 since the author did not show any direct interaction.
8. Line 344: consider changing 'We does not' to 'We do not'
Reference
Grosset AA, Loayza-Vega K, Adam-Granger É, Birlea M, Gilks B, Nguyen B, Soucy G, Tran-Thanh D, Albadine R, Trudel D. Hematoxylin and Eosin Counterstaining Protocol for Immunohistochemistry Interpretation and Diagnosis. Appl Immunohistochem Mol Morphol. 2019 Aug;27(7):558-563. doi: 10.1097/PAI.0000000000000626. PMID: 29271792.

Additional comments

I have made minor language edits in attached PDF file.

Reviewer 2 ·

Basic reporting

4. The English language should be improved significantly so that the international audience would have no misunderstandings over the text. There are way too many grammar errors, broken sentences, wrong usage of wordings, and inconsistent tense usage. Please emphasize on this part.
5. I suggest the author create an abbreviation list. It is very hard to follow the article at first before finding out what all the abbreviations mean.
6. More literature on the relationship between ETBR/ETAR and ICAM-1 needs to be included. The introduction explains their individual functions and relationships to the potential disease but did not spend too much effort into the investigations on correlation by others.

Experimental design

No comment.

Validity of the findings

1. It is a known fact that ETBR/ETAR have a correlation to ICAM-1 dating back to papers published in 1997, if not earlier. It is not something new to the field and there has been a study by Y Hayasaki et al. revealing a positive correlation between the two. Please comment and explain why your findings are almost polar opposite to previous findings.
2. While one-way ANOVA is the standard method to determine inter/intra group correlation or lack thereof, it is sometimes limited by sample size variation. Although in your experiment the sample sizes are conveniently the same, could you also use Mann-Whitney U-test to better support your findings?
3. Please check table 2 numbers and specifically the decimal places used. Are your measurements and percentages as accurate as what your decimal places suggest?

Additional comments

Please see attachment.

Annotated reviews are not available for download in order to protect the identity of reviewers who chose to remain anonymous.

---

## Round 0.2 · Minor Revisions

Please address the remaining issues pointed out by the reviewer and amend your manuscript accordingly.

·

Basic reporting

In this manuscript, the author showed that overexpression of ETbR in PCM-induced cardiomyositis rats improves cardiac function and reduces the inflammatory processes. The author then suggested that ETbR overexpression mediates its anti-inflamatory effect by suppressing ICAM-1 expression in cardiomyocytes, thus reducing immune cell recruitement to cardiac tissue.
The literature review is improved and sufficient at this point, in my opinion.
The figures' fonts are still distorted, specifically in Figures 2 and 3. This should be addressed before publication.

Overall, English in the revised manuscript is much improved, and the discussion section is also much easier to follow.

Experimental design

The experiment design is logical and easy to understand. I am currently satisfied with the method section.

Validity of the findings

The finding of this manuscript is exciting and logically valid at this point.
The data that support the idea that ETbR overexpression helps lessen cardiac inflammation and improves cardiac function is convincing.

The only major concern that I have left is the connection between ETbR overexpression and ICAM-1 downregulation. The only evidence that the author showed this anti-correlation relationship is the western blot in Figure 8. However, in that same western blot, the author did not include ETbR band within the same blot. To address this point, can the author show the whole blot, with ETbR, ICAM-1, and GAPDH with protein ladder with this manuscript, so that the audience can be easily convinced that the anticorrelation between ETbR and ICAM-1 is real?

Additional comments

1. Line 84: ‘vasorelaxation’ is not a common term. Please consider changing it to ‘vasodilation.’

---

## Round 0.3 · accepted · Accept

All remaining concerns of the reviewers are adequately addressed and the revised manuscript is acceptable now.